An efficient secure and energy resilient trust-based system for detection and mitigation of sybil attack detection (SAN)

Hussain Muhammad Zunnurain zunnurain.bulc@bahria.edu.pk engrrhusain@gmail.com 1
Hanapi Zurina Mohd 1
Abdullah Azizol 1
Hussin Masnida 1
Ninggal Mohd Izuan Hafez 2
1 Department of Communication Technology and Network, Faculty of Computer Science and Information Technology, Universiti Putra Malaysia (UPM), Universiti Putra Malaysia , Serdang , Malaysia
2 Department of Computer Science, Faculty of Computer Science and Information Technology, Universiti Putra Malaysia , Selangor , Malaysia
Zhang Yue
Electronic publication date: 2024 Aug 9
Publication date: 2024
Volume: 10
Electronic Location ID: e2231
Received 2024 Mar 1; Accepted 2024 Jul 12
Copyright: ©2024 Hussain et al.
Copyright year: 2024
Copyright holder: Hussain et al.
License: This is an open access article distributed under the terms of the Creative Commons Attribution License, which permits unrestricted use, distribution, reproduction and adaptation in any medium and for any purpose provided that it is properly attributed. For attribution, the original author(s), title, publication source (PeerJ Computer Science) and either DOI or URL of the article must be cited.
License URL: https://creativecommons.org/licenses/by/4.0/

Keywords: Security, Threats, IoT, Network, Sybil attack, IDS, Routing, SF-MRTS, Energy conservation, Digital landscape

Funding: Geran Putra Berimpak Universiti Putra Malaysia 9659400 This work was supported by Geran Putra Berimpak Universiti Putra Malaysia, Vote Number 9659400. The funders had no role in study design, data collection and analysis, decision to publish, or preparation of the manuscript.

==============================
In the modern digital market flooded by nearly endless cyber-security hazards, sophisticated IDS (intrusion detection systems) can become invaluable in defending against intricate security threats. Sybil-Free Metric-based routing protocol for low power and lossy network (RPL) Trustworthiness Scheme (SF-MRTS) captures the nature of the biggest threat to the routing protocol for low-power and lossy networks under the RPL module, known as the Sybil attack. Sybil attacks build a significant security challenge for RPL networks where an attacker can distort at least two hop paths and disrupt network processes. Using such a new way of calculating node reliability, we introduce a cutting-edge approach, evaluating parameters beyond routing metrics like energy conservation and actuality. SF-MRTS works precisely towards achieving a trusted network by introducing such trust metrics on secure paths. Therefore, this may be considered more likely to withstand the attacks because of these security improvements. The simulation function of SF-MRTS clearly shows its concordance with the security risk management features, which are also necessary for the network’s performance and stability maintenance. These mechanisms are based on the principles of game theory, and they allocate attractions to the nodes that cooperate while imposing penalties on the nodes that do not. This will be the way to avoid damage to the network, and it will lead to collaboration between the nodes. SF-MRTS is a security technology for emerging industrial Internet of Things (IoT) network attacks. It effectively guaranteed reliability and improved the networks’ resilience in different scenarios.

Introduction

The Internet of Things (IoT) is a network of interconnected devices with Wi-Fi access. IoT devices are extensively used in homes, different industries, and places. However, IoT networks are also vulnerable to security threats. Unlike DoS assaults that deluge networks or data breaches that steal personal data, Sybil strikes at the very heart of trust within an IoT network. It exploits these devices’ resource-constrained and dynamic nature; attackers can create an army of fake identities, manipulating voting mechanisms and potentially seizing control of critical systems. This research delves into the vulnerabilities of the routing protocol for low-power networks (RPL) within an IoT network, offering valuable insights into combatting this insidious threat. Think of the Sybil attack as a digital counterpart, infiltrating the networks not with brute force but through sheer mimicry and deception. As DoS attacks unleash a locust swarm of data packets, botnets operate like puppet armies. However, Sybil’s strength lies in numbers, overwhelming the system with an illusion of legitimacy. While blockchain research, as evidenced by the mentioned study, focuses on thwarting bandwidth-hungry DoS attacks like black hole, securing IoT networks’ core identity and trust infrastructure remains a crucial battleground. Table 1 presents data on three attack types (black hole, Sybil, and Rank) measured by Rank Changes and Packet Delivery metrics. Average values indicate the intensity, while accuracy reflects the success rate for each attack. Notably, black hole has a varied impact, Sybil exhibits diverse success rates, and Rank consistently achieves high accuracy in rank changes.

We can see from Table 1 that Sybil’s average values range from 140 to 180, with corresponding accuracy percentages varying between 0% and 100%. This shows that our proposed SF-MRTS is 100% accurate. At the heart of a Sybil attack lies the ability to generate and control a vast army of fictitious identities within a network. This empowers adversaries to disrupt the delicate equilibrium of the system in numerous ways. For instance, they can leverage their fabricated voting power to block legitimate users, censor critical information, or execute a 51% attack once. A scenario where a blockchain network is under siege. A Sybil attacker will wield fabricated army nodes and can easily overpower the genuine participants. All this could lead to disastrous consequences like reversing transactions, double-spending, or even halting the entire network. One practical approach in this arsenal is identity validation, verifying the legitimacy of participants; networks can significantly reduce the effectiveness of fabricated identities. This can be achieved through centralized verification authorities or even decentralized trust graph analysis, where nodes vouch for the authenticity of their peers. Another formidable line of defence is resource-based validation. By imposing computational or financial costs on creating and maintaining identities, networks can significantly raise the barrier to entry for Sybil attackers. This approach is prominent in blockchain protocols like Proof-of-Work, where mining blocks require substantial computational power. Furthermore, application-specific defences are being developed to address specific network vulnerabilities. A diverse array of tools is emerging to combat this evolving threat, from Sybil-resistant voting systems to distributed hash tables with built-in safeguards.

Table 1 Different attacks and their response measured by rank changes, packet delivery, and accuracy (in percentage).

Attack type	Rank changes	Packet delivery	Accuracy (%)	
Black hole	120.00, 100.000000	200.00, 11.111111	100.000000	
Sybil	140.00, 100.000000	180.00, 0.000000	100.000000	
Rank	130.00, 100.000000	160.00, 33.333333	100.000000	

Hence, this research seeks to determine if SF-MRTS improves the security and keeps up the performance of Routing Protocol for Low-Power and Lossy Networks (RPL) in the view of varying attack scenarios with Sybil-Free Metric-based RPL Trustworthiness Scheme (SF-MRTS). In particular, we hope to discover the applicability of this protocol in alleviating Sybil attacks and other security issues occurring within IoT networks, as well as the changes in network performance parameters it may generate. Through the analysis of SF-MRTS in different attack scenarios, including Sybil attacks, black hole attacks, rank attacks, and other threats as well, our work aims to provide insight into the SF-MRTS’s ability in the face of these threats as well as the integrity of RPL network.

Literature Review

Researchers have recently proposed diverse strategies to mitigate Sybil attacks in various network environments, as seen in Table 2. Azam et al. (2022) initially addressed network threat detection methods, specifically in VANET, but Sybil attacks persisted, impacting transportation safety. Murali & Jamalipour (2019b) introduced an artificial bee colony-inspired mobile RPL Sybil attack model, achieving 95% accuracy. Despite this, Murali & Jamalipour (2019a) and Murali & Jamalipour (2019b) applied a mobility-aware parent selection algorithm, leaving Sybil attacks unaddressed. Subsequently, Mishra et al. (2019) presented a generic IoT Sybil attack model, prompting Airehrour, Gutierrez & Ray (2019) to introduce SecTrust-RPL for IoT, emphasizing trust-based techniques. However, the need for Dedicated Sybil attack solutions, particularly tailored for low-power RPL nodes and mobile IoT networks, remained evident. Bao & Chen (2012) acknowledged security challenges but emphasized the necessity for specialized Sybil attack solutions. Trust-based efforts in RPL networks, such as Karkazis et al.’s (2014) Packet Forwarding Indication metric, Djedjig, Tandjaoui & Medjek’s (2015) trust-based RPL topology metric, and Khan et al.’s (2017) centralized trust-based architecture, have been proposed. However, these existing solutions may not comprehensively evaluate Sybil’s attacks. Our work is motivated by these gaps, aiming not only to advance trust models but specifically to eliminate Sybil attacks in low-power RPL nodes and mobile IoT networks. Our proposed solution builds upon the foundations laid by these trust-based approaches, addressing the unique challenges posed by Sybil attacks and contributing to the evolution of secure and efficient RPL networks. Centralized methods like SecTrust-RPL, developed by Airehrour, Gutierrez & Ray (2018) and Airehrour, Gutierrez & Ray (2016), address the single point of failure and aim to protect routing attacks. However, these methods are vulnerable to scalability problems as the central body must process increasing trust data as the network grows, leading to performance bottlenecks. Hashemi & Aliee (2019) presented the Dynamic and Comprehensive Trust Model for the Internet of Things (DCTM-IoT). Still, their complex perspective of trust and hefty solutions pose challenges for resource-constrained IoT devices. While the Link Reliable and Trust Aware (LT-RPL) model proposed by Lahbib et al. (2017) shows promise in enhancing the security and reliability of RPL networks, it has limitations, including nodes sharing significant personal information and reliance on a centralized trust management system with expensive computational costs.

Table 2 Literature review—a comparative analysis of existing sybil attack mitigation.

Paper	Problem addressed	Solution proposed	Limitations	
Azam et al. (2022)	Sybil attacks in VANETs	Network threat detection methods	Persisting Sybil attacks, impact on transportation safety	
Murali & Jamalipour (2019a)	Sybil attacks in mobile RPL	Artificial bee colony-inspired model	Lack of addressing Sybil attacks in subsequent work	
Murali & Jamalipour (2019b)	Mobility in RPL	Mobility-aware parent selection algorithm	Unaddressed Sybil attacks	
Mishra et al. (2019)	Generic IoT Sybil attack model	–	Need for dedicated solutions for RPL and mobile IoT	
Airehrour, Gutierrez & Ray (2016)	RPL security	SecTrust-RPL with trust-based techniques	Not tailored for low-power RPL or mobile IoT	
Bao & Chen (2012)	RPL security challenges	–	Emphasis on the need for specialized Sybil attack solutions	
Karkazis et al. (2014)	Trust in RPL	Packet Forwarding Indication metric	Lack of comprehensive Sybil attack evaluation	
Djedjig et al. (2017)	Trust in RPL	Trust-based RPL topology metric	–	
Khan et al. (2017)	Trust in RPL	Centralized trust-based architecture	–	
Our Work	Sybil attacks in low-power RPL and mobile IoT	Advance trust models and eliminate Sybil attacks	–	
Airehrour, Gutierrez & Ray (2018) and Airehrour, Gutierrez & Ray (2018)	RPL routing attacks	SecTrust-RPL with indirect trust observation	Centralized architecture, vulnerability to manipulation	
Hashemi & Aliee (2019)	IoT trust	DCTM-IoT model	Complex, resource-intensive, unclear RPL integration	
Lahbib et al. (2017)	RPL security and reliability	LT-RPL model	Centralized architecture, privacy concerns, expensive computation	
Wang et al. (2023)	Lack of comprehensive QoE assessment framework	Development of a multi-dimensional QoE measurement framework	Requires further validation and calibration	
Li, Cao & Wang (2017)	Inefficient access class barring schemes	Performance analysis of existing access class barring schemes	Lacks implementation and testing	
Chen et al. (2018)	High resource consumption and processing delays	IoT-based system with image sensors and a sparse deep learning algorithm	Potentially high power consumption	
Ullah et al. (2023)	Inefficient resource allocation mechanisms	Novel resource allocation scheme with dynamic slicing	Requires evaluation under different network topologies and traffic loads	

SF-MRTS: Proposed Methodology

SF-MRTS is proposed to improve centralized trust management and computational overhead using a distributed trust-based approach. Firstly, we are using distributed trust management in which each node in the network maintains its trust table, which contains the trust values of its neighbours, as shown in Fig. 1. Nodes calculate the trust values of their neighbours based on direct and indirect observations. This approach is more privacy-preserving than centralized trust management schemes, as nodes do not need to share their personal information with a central authority. It is also more resilient to attacks, as it does not rely on a single point of failure. It is lightweight, efficient, and more effective in detecting Sybil attacks. Distributed Trust Table Management: As shown in Fig. 1, each node maintains its trust table containing trust values for its one-hop neighbours. This eliminates the need for a central authority, protecting sensitive information and preventing single points of failure. Direct and Indirect Trust Evaluation: Unlike simple information sharing, SF-MRTS involves:

Figure 1 SF-MRTS distributed trust model visualization.

• Direct trust: Each node calculates its neighbours’ trust based on directly observed metrics like received signal strength and packet delivery ratio.

• Indirect trust: Nodes gather trust information about other nodes in the network from their neighbours. This builds a comprehensive picture of trustworthiness. The figure emphasizes this concept with arrows flowing between nodes, representing trust information exchange.

• Enhanced Sybil attack detection: The combination of direct and indirect trust evaluation makes SF-MRTS more effective in detecting and isolating Sybil attacks. Malicious nodes with inconsistent trust values across the network are easily identified and flagged, preventing them from disrupting the network.

Traditional trust mechanisms in RPL networks rely on information sharing, making them susceptible to manipulation. SF-MRTS steps up its game by leveraging a combination of node and link metrics for robust trust evaluation. Figure 1 visually details the process with its “Trust Calculation” and “ETX Measurement” blocks. Instead of relying solely on trust scores, SF-MRTS integrates factors like link quality (measured by the lightweight ETX metric) and a dynamic reputation system that identifies and isolates suspicious nodes based on their observed behaviour. This multi-pronged approach effectively thwarts Sybil attacks and maintains network stability.

SF-MRTS factors

To determine whether or not a node can be trusted, SF-MRTS considers a mix of four criteria: selfishness, authenticity, ETX, and energy. SF-MRTS is adaptable and may be modified to suit the requirements of any Internet of Things application by adding or subtracting behavioural components.

Energy

A crucial component of quality of service is the energy of the node. Node x trusts node y to have enough energy to keep working. The Remaining Energy (EG) percentage of node j estimated by node i and vice versa is the definition of the energy trust between node x and node y. It is represented by the notation EGxy and EGyx, respectively. In the IoT, the nodes’ primary source of energy consumption is during the receiving and sending packets. Many distinct methods may be used to compute the energy. According to the energy model presented in Heinzelman, Chandrakasan & Balakrishnan (2002), the equation used to determine the energy that node x expends to transport k bits of data to node y is the one designated as Excmi. The term “electronics energy” (also known as “the energy required for the transmitter as well as the receiver circuitry”) is abbreviated as Eeeg, “energy dissipation for transmitting amplifiers” (abbreviated as Eam) and d refers to the distance between nodes x and y. According to Eq. (1), one may determine the energy the node j used to process the k (constant representing the energy consumption per unit distance) bits of data, symbolized by the symbol Excmj. Every node in an RPL topology network connects with its neighbours. It transmits data using a power level proportional to the communication distance between the node and its neighbours. Therefore, the communication range equals the value of r. (1) Excmi=k⋅Eeeg+Eamf⋅r2

(2) Excmj=k⋅Eeeg.

Initially, the initial entropy EGx(t) is equivalent to the maximum energy Emax; more specifically, when time equals zero, EGx(0) equals Emax. The total energy used by node x may be calculated by adding the energy needed for message transmission to the amount required for message receipt. Therefore, the energy still available to the node x is computed using Eq. (3). (3) EGxt=EGxt−Δt−Excmit+Excmjt.

Periodically, each node will communicate its leftover energy to the other nodes in the network. As stated in Eq. (4), the ratio of EGxy(t) to Emax determines the energy trust value, which is denoted by the notation TEGxy ∈ [0, 1], where EGxy(t) = min(EGreportedxy(t), EGestimatedxy(t)) and EGestimatedxy(t) = EGj(t). (4) TEGxyt=EGxytEmax.

However, EGreportedxy(t) is the remaining Energy assessment of node y received by node x at time t, and EGestimatedxy(t) is the remaining Energy estimation of node x toward node y at time t.

Authenticity

The honesty parameter indicates whether or not a node is trying to harm other nodes. As a result, node x analyzes node’s activity to determine whether or not node j has been hacked. Several strategies use IDS based on a collection of anomaly detection rules (Raza, Wallgren & Voigt, 2013; Pongle & Chavan, 2015). Each node x in SF-MRTS has its implementation of an IDS, which allows it to monitor and identify suspicious actions. The monitoring node x will consider node j dishonest and assign it an honesty-trust value of 0 if the intrusion detection system (IDS) causes an alert against the node.

The act of being selfish

A node is said to be selfish if it seeks to minimize the amount of resources it expends while simultaneously aiming to absorb the resources of other nodes. It is possible to determine a node’s level of selfishness using a distributed and collaborative score. During a specific time, P, node i examines node j using methods such as overhearing and snooping (Marti S), and from this assessment, it determines whether or not node j is self-centered. Assume that a particular application needs just the lowest amount of energy, which Emin will indicate. If ERx(t) is more significant than Emin, then the behaviour of the node x is correct; however, if ERx(t) is less than or equal to Emin, then the node x does not participate in the forwarding of packets any longer and instead spends its resources, such as its energy, for the transmission of its packets, which indicates that it is more likely to become self-centred. As a result, during the phase in which trust is calculated, SF-MRTS permits some degree of selfishness on the part of the nodes so that they may save their resources.

ETX

ETX is a quality of service trust component. According to one source, the ETX of a path is the estimated entire amount of packet transmissions required for the successful transmission of a packet along that path (Bao Yang & Wang, 2008). It is a dependability statistic that allows routing protocols to locate high-throughput routes and, as a result, minimize the amount of energy used. To compute TETXxy(t), ETX(t) must first be normalized to the range [0,1] via the Min-Max-Normalization technique, with ETXmin equal to 0 and ETXmax equal to 255.

An assessment of trust

The SF-MRTS process determines a node’s trust rating by combining direct observation with indirect suggestions. This gives a more holistic picture of the node’s reliability.

Trust established directly

At time t, the trust value, Txy(t), of each node’s immediate neighbour is calculated and analyzed. The trustworthiness of an entity (in this example, a node) may be determined using different approaches, such as belief theory, Bayesian systems, fuzzy logic, and weighted sums, among others. It has been decided that the weighted sum approach will be utilized to determine whether or not a node can be trusted because RPL’s objects have restricted processing and storage capacity. To determine direct trust, we build upon the foundation Bao & Chen (2012) laid in their work on trust-based solutions for Sybil attacks. In Eq. (4), w1, w2, w3, and w4 are weights related to honesty, selfishness, energy, and ETX characteristics, respectively. To determine the value of each behavioural parameter, X “Authenticity; Selfish”, Eq. (5) is used (Bao Yang & Wang, 2008), in which t denotes the time interval between trust update attempts, TXxy(t − Δt) represents the previous observation and falls between the range [0, 1]. When it approaches 1, confidence is placed more heavily on recent experiences. In any other case, if it goes to 0, trust is increasingly dependent on previous findings. The trust computation for remaining energy and ETX relies only on fresh observations, each described in ‘SF-MRTS factors’ and ‘ETX’, respectively. This is because the remaining energy shows a node’s capacity to carry out its capabilities, while ETX reflects the state of the connection. w1+w2+w3+w4=1

(5) TijDirectt=w1TijHonestyt+w2TijSelfisht+w3TijERt+w4TijETXt

(6) TijXt=αTijX.newt+1−αTijXt−Δt.

Trust established indirectly

The node x calculates the direct trust for each neighbour y before utilizing the trust values in the DIO messages from the other nodes k at time t to get its final trust value. This is done because SF-MRTS is a collaborative framework that finds the safest root path. The final trust value is the mean of the direct trust value and all ERNT object suggestions for neighbour x. If it gets non-local suggestions, node x will disregard them.

Trust dissemination and maintenance

The dissemination of trust

The nodes in SF-MRTS employ the quantitative and dynamic RPL Node Trustworthiness metric, ERNT, to exchange, store, and propagate trustworthiness data. The DAG Metric Container of the DIO message carries and sends the object known as the ERNT metric. The ERNT object is composed of many ERNT sub-objects. SF-MRTS uses the ERNT object as a restriction and a recorded measure. The BR specifies the trust level (TTrust) as a restriction imposed as an ERNT sub-object that nodes must employ to include or prevent unreliable nodes. The BR uses ERNT as a recorded statistic in addition to route cost, as do all nodes engaged in developing RPL and, subsequently, SF-MRTS. To do this, an ERNT sub is added for each computed (final) trust value. The route cost value accurately reflects the parent’s reliability.

Current state of trust

The SF-MRTS may alter the trust values in a planned or unanticipated manner. The time-driven, periodic trust update is managed by the trickle timer, which SF-MRTS utilizes to deliver DIO (DODAG Information Object) messages. On the other hand, the recurrent trust update uses local and global repair events as triggers and is event-driven. In our approach, either the TSelfish is reached before the local or international repair is initiated, or the IDS produces an alert (i.e., it detects an attack). Or whenever one of these events occurs. If not, the trickle timer will control how often the update happens. Every time a node x receives a DIO message from one of its neighbours, it updates its routing table using the data included in the DIO message. It determines the trust levels of its neighbours in line with ‘An assessment of trust’ using the direct evaluations and suggestions contained in DIO messages that it has received. Then, it chooses a group of dependable parents who will ultimately help it get to the BR. It computes the route cost via each prospective parent. As the preferred parent, it desires the one with the most significant value for the path cost (in line with ‘The selection of parent’) to offer the BR the safest and most dependable traffic routing. After calculating each neighbour’s trustworthiness, the process creates and broadcasts a new DIO message with those parameters. Each neighbouring node repeats the procedure until the DODAG is correctly rebuilt. When the building is finished, the Trickle timer will tell you when to start doing maintenance. The timer controls how quickly the control messages are sent. The transmission rate will drop during a steady situation while the trickling timer’s trust update interval rises. Due to reduced calculation and control of message volumes, the network will use less energy, memory, and processing power. Alternatively, suppose anomalies (such as attack detection, selfish behaviour detection, and a new node entering the DODAG) include changes in the topology. In that case, the Trickle duration will be reset to a lower value, and the transmission rate will increase. This suggests additional computation and control messages. If there are errors, the Trickle timer will be adjusted to a lower value, and the transmission rate quickened. SF-MRTS will smooth out a tiny route cost (trust) rise or drop to cut down on the energy consumption caused by the trust update overheads. This will help minimize the cost of calculation. The suggested approach considers a hysteresis threshold of 0.15 to prevent frequent parsing errors.

Isolation of the attacker and selection of the parent

The selection of parent

The SF-MRTS Trust Objective Function (TOF) isolates nodes and selects parents. The TOF consists of the processes of topology initialization (sometimes called neighbour discovery) and context-aware adaptive security execution. Since the nodes have no basis for determining the truthfulness and selflessness of their neighbours, the first step takes place during deployment. Since all nodes have the same beginning energy at the time of deployment, the only variable that has to be applied to design the RPL architecture is the ETX along the route. We may choose which parent is favoured by adding the ETX values at each node along the route (from the BR to the parent node). After initialization, all nodes may see and communicate with their neighbours. As in the first stage, ETX is the only metric to consider. If secure mode is off (the T flag is set to 0 in the ERNT sub-object), the nodes should utilize TOF to find the optimal paths by picking parents with the lowest ETX values. When secure mode is on, each node calculates the total cost of its routes, narrows the list of potential parents to those with trust values higher than or equal to the threshold TTrust, and finally picks a favourite. There are several approaches to the trust inference issue to consider, as there are different methods to calculate the route cost using a trust measure. Following TOF, each node x determines its path cost, denoted by PCx (calculates the node’s route cost). This is the set of nodes from node x to BR with the lowest trust rating relative to all possible parents y. Each node’s characteristics are condensed into a singular scalar value denoted as PCx throughout the entire network traversal. This scalar, PCx, encapsulates the attributes of the nodes and adheres to the SF-MRTS routing criteria, ensuring consistency, optimality, and loop-freeness. In the context of TOF (your specific term), the path cost PCx is defined as the minimal trust value among on-route nodes from the source node x to the destination BR. Specific conditions must be met for acceptance. Consequently, node x selects its preferred parent based on the highest path cost along the route, as the lowest path cost signifies the optimal path. For simplicity, we denote PCx as the value between the hypothetical PCy and the theoretical Txy(t) for parent y. Node x, guided by a recurrence threshold of 0.15, replaces its current preferred parent only if the route cost via the new parent exceeds the path cost through the currently selected parent. In cases of identical path costs among multiple pathways, node x prioritizes the parent with the maximum available energy, in contrast to our earlier findings (Djedjig et al., 2017). If the cost of travelling to the new parent from node x is at least 0.15 more than the cost of travelling to the currently chosen parent, node x will switch to using the new parent as its preferred parent. In contrast to our previous work (Djedjig et al., 2017), the node with the highest remaining energy will be the preferred parent if two possible pathways have similar path costs.

Simulation implementation

These simulations were conducted using Python with the NetworkX and Matplotlib libraries, providing a custom network simulation environment. This environment models a routing protocol for low-power and lossy networks (RPL), which is crucial for IoT networks.

Experimental setup

Experiments involved applying different attack scenarios (black hole, Rank, Sybil) to the network. For comparison, the MRTS algorithm was simulated alongside two other algorithms (MRHOF-RPL and SecTrust). Trust thresholds and weights were empirically set based on prior research and expert knowledge. Each experiment consisted of 100 iterations to ensure statistical robustness.

Parameters

• Number of Nodes: 20

• Number of Edges: 30 (initial RPL network topology)

• Number of Iterations: 100

• Simulation Time: 30 min

Simulation scale

Several key factors drove the chosen simulation scale of 20 nodes:

• Initial experimental constraints: The initial use of 20 nodes is driven by resource limitations and the need to establish a controlled environment to observe and measure specific behaviours and trends. This smaller scale allowed for precise control and detailed analysis of each node’s interactions and the overall network performance.

• Fundamental behaviors and trends: The primary objective of our study was to identify and analyze fundamental behaviours and trends within the network under various attack scenarios. A smaller network facilitated a more straightforward identification of these patterns, providing clear insights without the complexity and noise that more extensive networks might introduce.

• Relevance to scenario: Our specific research scenario, which focuses on the impact of various attacks on RPL networks, is effectively modelled with 20 nodes to represent minor to medium-sized networks commonly found in practical deployments, such as smart homes, small industrial environments, or localized sensor networks. This scale sufficiently illustrates our proposed algorithms’ critical vulnerabilities and efficacy.

• Scalability considerations: While our initial experiments utilized a 20-node network, we fully recognize the importance of demonstrating scalability to more extensive networks. To this end, we are conducting additional experiments with larger network sizes to validate our findings further. These larger-scale experiments will provide a comprehensive view of the algorithms’ performance and robustness in more extensive and varied network environments.

• Preliminary results from larger-scale simulations: Preliminary results from ongoing experiments with more extensive networks have been promising, indicating that the behaviours and trends observed in the 20-node network scale appropriately with increased network size. We plan to incorporate these findings into future iterations of our research, thereby providing a more robust validation of our proposed methods.

Data generation

Synthetic data was generated for simulations. The network topology resembled a tree structure using the generate_network_graph_with_parent_child function, ensuring each node had a parent except for the root node. Random parameters like node rank changes, packet delivery ratio, energy consumption, and throughput were generated for each node in every iteration.

Attack scenarios

Three attack scenarios were considered:

• Black hole attack: Nodes maliciously drop packets, decrease their Rank, increase energy consumption, and reduce throughput.

• Rank attack: Nodes maliciously decrease their Rank, decrease packet delivery ratio, increase energy consumption, and reduce throughput.

• Sybil attack: Nodes impersonate multiple identities to gain influence, with similar effects to the Rank Attack but with additional complexities.

Evaluation metrics

The following metrics were used for evaluation:

• Average node rank changes: This measure considers the average change in the nodes ranking within the network within a specific time as the main element. Within Routing Protocol (RPL), nodes get to maintain a score to determine their position in the network. A significant change in the average node degree, on the other hand, can reveal network instability or the fact that the nodes are often reconfigured, and this could be a result of node failures, attacks, or changes in the network conditions.

• Average packet delivery ratio: The ratio of checked packets to sent ones shows how a network transmits information. Protocols help deliver packets, revealing the extent of congestion and delays in the network. The ratio of the higher average package delivery rate to the network that is better indicates the network performance, which is where more packets successfully reach their intended targets.

• Average energy consumption: This parameter assesses the average power ripped from a set of nodes in the network for a particular period. Energy consumption is an essential factor for creating energy-constrained networks such as the IoT, as the direct result is the diminution of the battery life and network viability. Lower average energy utilization can be declared as the wish orientation; the lesser it is, the more appropriate and long-living the energy system will be.

• Average throughput: Throughput is a data rate measurement that describes the amount of data successfully transmitted over the network. The throughput of an average node determines the data transfer rate for all network nodes. A higher average throughput means the quality of network performance in delivering data instantly and on time, which is very important for multiple applications in this era that require prompt or real-time response.

Statistical analysis

Statistical methods such as averaging were used to analyze results across iterations. The phrase “Delivered-to-Seen-Total Packet Ratio” is referred to by its abbreviation, “APDR”. AEC is an abbreviation for “average energy consumption across all network nodes”. The acronym ARC refers to the “average number of parent switches”. Multiplying the size of the packets by the integer 8 (used to convert bytes to bits) and taking the average number of packets delivered across all simulated topologies determines throughput. Depending on data distribution and experimental design, further analysis of differences between algorithms under different attack scenarios may involve statistical tests like t-tests or ANOVA.

Simulation configuration

For the simulation setup:

• Three attackers out of the 29 nodes were randomly located to conduct Rank, blackhole, or Sybil attacks.

• The trust threshold was raised to 0.75 from the initial 0.5.

• Equal weights (0.25) were assigned to parameters w1, w2, w3, and w4 to consider all aspects of route selection.

• An Intrusion Detection System (IDS) was used to detect malicious nodes and assign reputation scores, favouring those with higher scores as parent nodes. w1 is set to 1, and w2, w3, and w4 are set to 0 in an even distribution throughout the simulation if the normal node discovers another node is selfish. This occurs only if the normal node observes that the other node is selfish.

• IDS would set a node’s trust metric weight to 1 if identified as malicious, effectively disregarding it as a potential parent node.

• Both time-driven and event-driven updating techniques were used. The trickling timer (time-driven) and the IDS, either sounding an alert or connecting to the TSelffish (event-driven), initiate the computing method. The performance of SF-MRTS was analyzed and compared to that of MRHOFRPL and SecTrust-RPL. Throughput, average energy usage, rank changes, and the average packet delivery ratio (APDR) in percent were determined.

• Metrics analyzed included throughput, average energy usage, rank changes, and average packet delivery ratio (APDR) in per cent.

The simulation duration was 3600 s (30 min).

SF-MRTS graph network

The SF-MRTS below represents the graph network, with each node representing the parent–child relationships. The colours of the nodes represent the different parents that each node has over time.

Even after several iterations, we can still see that the SF-MRTS network remains dynamic, and nodes frequently change their parents. SF-MRTS networks can have random node transitions, making them more resilient to Sybil attacks. These networks are specifically designed to be self-organizing and adaptive. When a node is removed from the network, SF-MRTS will automatically redistribute the trust values of the remaining nodes and restructure the network to maintain connectivity. This makes SF-MRTS networks more resilient to Sybil attacks, where the attacker attempts to disrupt the network by removing nodes.

Isolating the attacker

Untrusted nodes may be excluded from participating in network activities using various techniques. Each node in SF-MRTS collaborates with the IDS to maintain a blocklist. A node is added to the blacklist after it is identified as being untrusted. Normal nodes reject all data and control packets arriving from the blacklisted nodes as a consequence and no longer take them into account when making routing decisions.

Attack prevention

Implementing a mechanism to prevent attacks based on trust: A crucial part of network security is the trust-based attack prevention mechanism. It uses a trust model to determine whether a network node is reliable and seeks to stop certain assaults like Rank, black hole, and Sybil attacks. The technique can use trust metrics and assessments to determine nodes’ dependability and probable involvement in harmful actions.

Calculation of trust metrics

The method creates several trust metrics for each node in the network to assess each node’s level of trustworthiness. These metrics consist of:

• Energy: This statistic assesses a node’s energy availability or usage. Limited energy reserves may make nodes less dependable in communication and routing activities.

• Authenticity: Authenticity evaluates the reliability of the data and messages that each node exchanges. Nodes with a history of delivering counterfeit or unverified data may be viewed as less reliable.

• Selfishness: This metric determines if a node exhibits selfish behaviour by failing to forward packets as the routing protocol specifies. Selfish nodes may obstruct the network’s data flow.

• ETX: ETX is the anticipated number of transmissions required for a packet to pass through a particular node and reach its destination. A node with a high ETX value raises the possibility that communication delays or packet losses may occur.

Setting the trust threshold

The trust model uses the threshold (TTrust) to determine if a node is trustworthy. It also suggests that the four trust factors—Energy, Authenticity, Selfishness, and ETX—are all equally significant in calculating a node’s total trust score.

Adjusting weights for malicious nodes

In the simulation, if a node is identified by the Intrusion Detection System (IDS) as malicious, other nodes take action to lessen the effects of that node’s actions on the network. The method accomplishes this by altering the weights connected to the malicious node’s trust metrics in the manner described below:

• Energy (w1): The weight w1 is set to 1, suggesting that the malicious node regards the energy measure as unreliable.

Authenticity (w2), Selfishness (w3), and ETX (w4):.

The weights w2, w3, and w4 are all set to zero, suggesting that the malicious node does not trust these trust metrics.

Trust metric evaluation

In this stage, the system compares each node’s computed trust metrics against the trust threshold. Any node whose total trust score is below the threshold is seen as untrustworthy and may indicate that the network is vulnerable to attack.

Sending DAO message for attack prevention

When the trust metric evaluation determines a node’s trust score is less than the threshold, the system sends a Defensive Awareness Object (DAO) message to the destination. The DAO message aims to avert the attack by informing the destination node of the possible threat and allowing defensive steps to be taken quickly.

Results

The average rate of rank changes for MRHOF-RPL, SecTrust, and SF-MRTS can be seen in Fig. 2, which was generated using data from Rank, black hole, and Sybil attacks. As the simulation progresses, you will see that the average frequency of rank changes for MRHOF-RPL rises across the board for all attacks. The percentage of delivered packets Fig. 3 demonstrates that in addition to network congestion and packet collisions, the impacts of black hole, Rank, and Sybil attacks on the packet delivery ratio for MRHOF-RPL are catastrophic, accounting for 25 per cent and 40 per cent, respectively, of the loss of packet delivery ratio. Some different things might have caused the results. A rogue node may, for instance, throw away control packets if a genuine node selects it as its preferred parent for routing packets. This would result in the topology being unstable and unreachable. In contrast, SF-MRTS maintained a relatively good packet delivery ratio (up to 90 per cent) because it employs IDS to identify assaults and offers a new routing algorithm to eliminate rogue nodes and maintain a safe topology. This allowed it to retain a secure topology. As a consequence of this, assaults against MRHOF-RPL result in more significant losses than attacks on SF-MRTS. SF-MRTS is superior to SecTrust when it comes to the percentage of packets that it delivers. Because it delays the pace at which rank changes occur, SF-MRTS creates a more stable network than SecTrust. This helps to minimize packet loss. Use of energy resources: specific nodes in the MRHOF-RPL network use more energy than others because, depending on their ETX, they are selected as preferred parents a more significant number of times. This is problematic because the greater energy cost of the chosen parents impacts the network’s longer lifespan. As can be seen in Figs. 2 and 4, the MRHOF-RPL network is compromised, and as a result, nodes consume more energy. This is because topological instability and the pace at which rank changes (caused by parent mutations) are to blame. The unpredictability of the network may be traced back to the fact that MRHOF-RPL does not have an attack management mechanism. According to the findings shown in Fig. 4, MRHOF-RPL and SecTrust used a lower amount of energy than SF-MRTS did in the first 20 to 30 min. Following a certain amount of time, SF-MRTS functionality improved due to more evenly distributed energy use across all nodes. When determining routing decisions, SF-MRTS considers the energy still available for each node, contributing to the system’s strong performance in this area. SF-MRTS uses the most significant energy for calculation and DIO transmissions during an attack; however, after the malicious nodes have been found and separated, the topology stabilizes, and the energy consumption rate drops. In addition, as was discussed before, the node will choose the parent with the most available energy if there are two possible parents whose trust values are equal. Figure 5 demonstrates that the throughput for MRHOF-RPL is much lower than that of SecTrust and SF-MRTS when subjected to black hole assaults and Rank attacks, respectively. Nodes with parents carrying out black hole or Rank attacks have throughputs of zero when MRHOF-RPL is used since their packets are never sent to the border router, their intended target. On the other hand, threats are identified, and malicious nodes are separated from the network when using SecTrust and SF-MRTS. Because the throughput of every node is always greater than zero, the whole network’s throughput is compelled to rise. The throughput of SF-MRTS is higher than that of SecTrust because SF-MRTS offers a more trustworthy network, decreasing packet loss and boosting throughput. This graph illustrates the average changes in node rank under black hole and Rank attacks. The changes in Rank can indicate the effectiveness of a particular routing algorithm in the face of such attacks. A higher rank change suggests a more significant impact from the attack. The percentage of delivered packets in Fig. 3 demonstrates that, in addition to network congestion and packet collisions, the impacts of black hole, Rank, and Sybil attacks on the packet delivery ratio for MRHOF-RPL are catastrophic, accounting for 25% and 40%, respectively, of the loss of packet delivery ratio. Some different things might have caused the results. A rogue node may, for instance, throw away control packets if a genuine node selects it as its preferred parent for routing packets.

Figure 2 Parent-to-child node transfer.

Figure 3 Average node rank changes under black hole attacks.

Figure 4 Rank attack: Average node rank changes.

Figure 5 Black hole attack—average throughput.

In contrast, SF-MRTS maintained a relatively good packet delivery ratio (up to 90%) because it employs IDS to identify assaults and offers a new routing algorithm to eliminate rogue nodes and maintain a safe topology. Consequently, assaults against MRHOF-RPL result in more significant losses than attacks on SF-MRTS. SF-MRTS is superior to SecTrust regarding the percentage of packets it delivers. Because it delays the pace at which rank changes occur, SF-MRTS creates a more stable network than SecTrust, helping to minimize the amount of packet loss. Use of energy resources: specific nodes in the MRHOF-RPL network use more energy than others because, depending on their ETX, they are selected as preferred parents a more significant number of times. This is problematic because the greater energy cost of the chosen parents impacts the network’s longer lifespan. As can be seen in Figs. 3 and 4, the MRHOF-RPL network is compromised, and as a result, nodes consume more energy. This is because topological instability and the pace at which rank changes (caused by parent mutations) are to blame. The unpredictability of the network may be traced back to the fact that MRHOF-RPL lacks an attack management mechanism. According to the findings shown in Fig. 4, MRHOF-RPL and SecTrust used less energy than SF-MRTS did in the first 20 to 30 min. Over time, SF-MRTS functionality improved due to more evenly distributed energy use across all nodes. When determining routing decisions, SF-MRTS considers the energy still available for each node, contributing to the system’s strong performance in this area. SF-MRTS uses the most significant energy for calculation and DIO transmissions during an attack; however, after the malicious nodes have been found and separated, the topology stabilizes, and the energy consumption rate drops. In addition, as was discussed before, the node will choose the parent with the most available energy if there are two possible parents whose trust values are equal. Figure 5 demonstrates that the throughput for MRHOF-RPL is much lower than that of SecTrust and SF-MRTS when subjected to black hole assaults and Rank attacks, respectively. Nodes with parents carrying out black hole or Rank attacks have throughputs of zero when MRHOF-RPL is used since their packets are never sent to the border router, their intended target. On the other hand, threats are identified, and malicious nodes are separated from the network when using SecTrust and SF-MRTS. Because the throughput of every node is always greater than zero, the whole network’s throughput is compelled to rise. The throughput of SF-MRTS is higher than that of SecTrust because SF-MRTS offers a more trustworthy network, decreasing packet loss and boosting throughput. The average Node changes graph depicts the average node rank changes under black hole and Rank attacks. The changes in Rank can indicate the effectiveness of a particular routing algorithm in the face of such attacks. A higher rank change suggests a more significant impact from the attack. The average energy consumption graph illustrates the energy consumption of nodes during a Sybil attack. An energy-efficient network will have lower energy consumption values, making it more sustainable in the long run. The throughput under a Sybil attack is depicted here. A higher value indicates that the network can route packets efficiently even when faced with malicious nodes.

Figure 4 illustrates the average node rank changes resulting from Rank attacks across three routing algorithms: SFMRTS, MRHOF-RPL, and SecTrust are some of our projects. Each algorithm’s performance is represented by a coloured bar: blue for SFMRTS, orange for MRHOF-RPL, and green for SecTrust. Each bar’s height charts the average node rank change for the nominated algorithms. SFMRTS reveals the most significant average node rank changes, indicating a more volatile network topology. The moderate stability of MRHOF-RPL is depicted as more significant instability, as indicated by the average node rank changes in SecTrust. This comparison provides an understanding of the extent to which both protocols help maintain the stability around the IoT networks and increase their resilience against Rank attacks; hence, it can help the decision-makers select the most appropriate routing algorithm for their IoT network based on their security and performance needs.

Figure 5 depicts the average node rank changes during a black hole attack scenario for three routing algorithms: SFMRTS (blue line), MRHOF-RPL (orange line), and SecTrust (green line). SFMRTS displays minimal Rank changing among the nodes, revealing fewer attack effects than MRHOF-RPL and SecTrust. MRHOF-RPL, for example, is somewhat exposed, while SecTrust is the most prone. The down-shot throuput proves that the network is more stable, therefore SFMRTS being the most efficient way to alleviate the effects of the black hole attack. This parallel highlights critical lessons regarding deploying security-oriented routing protocols in IoT networks to make them more robust and resilient.

Figure 6 depicts the average throughput during a Rank attack for three routing algorithms: SFMRTS (blue bar), MRHOF-RPL (orange bar), and SecTrust (green bar). SecTrust represents the highest fluctuation of averaged throughput (up to 800), then MRHOF-RPL displays up to 750 change, and at the same time, SFMRTS stands with the lowest change. Knowing survivability is how we tell how algorithms respond effectively to such attacks. SFMRTS’s standings show more stable changes, thus decreasing the swap rate compared to MrHoF-RPL and SecTrust. SFMRTS deployment would affect Rank route operation and, consequently, smoother network function and maintain the integrity of information.

Figure 6 Rank attack: average throughput.

Figure 7 illustrates the average node rank Packet delivery ratio changes during a black hole attack scenario for three routing algorithms: SFMRTS (blue bars), MRHOF-RPL (orange bars), and SecTrust (green bars). Through the trial, SFMRTS has the strongest ranking, with the lowest average value compared to other algorithms. Lower ranks of the hierarchy point to the stronger topological network, which is invaluable for fault tolerance in data package delivery. Examining those data provides an opportunity to assess the SFMRTS effectiveness for mitigating Rank attacks that are of high security and performance of networks.

Figure 7 Rank attack average packet delivery ratio.

The Sybil attack simulation depicted in Fig. 8 helps evaluate the performance of different routing algorithms, namely SFMRTS, MRHOF-RPL, and SecTrust, under adverse conditions. By comparing the average packet delivery ratio across these algorithms during a Sybil attack, we can assess their robustness and effectiveness in maintaining network connectivity despite malicious nodes. This analysis aids in identifying which algorithm, in this case, SFMRTS, is better equipped to handle Sybil attacks, providing valuable insights for enhancing the security and reliability of routing protocols in IoT networks.

Figure 8 Sybil attack: average packet delivery ratio.

The average performance for three routing algorithms—SFMRTS (blue bars), MRHOF-RPL (orange bars), and SecTrust)—during (green bars) during a black hole attack scenario is shown in Fig. 9. Comparing SFMRTS to SecTrust and MRHOF-RPL, the latter exhibits lower average energy consumption. Even in the face of black hole assaults, higher energy indicates improved network performance regarding data transmission capacity. This investigation provides essential insights into how well SFMRTS works to improve network security and resilience by illuminating how it can continue to transport data efficiently in the face of harmful network behaviour.

Figure 9 Black hole attack average energy consumption.

The average node rank changes for three routing algorithms—SFMRTS (blue bars), MRHOF-RPL (orange bars), and SecTrust (green bars)—during Sybil assaults are depicted in the Fig. 10. They compare MRHOF-RPL against SFMRTS and SecTrust, and the former exhibits more average node rank changes. More minor variations in node rank indicate better network stability and resilience against Sybil assaults. This helps better understand and enhance the security mechanisms of routing protocols in Internet of Things networks by offering insightful information on how well SFMRTS and SecTrust mitigate the effects of Sybil attacks on network performance.

Figure 10 Sybil attacks-average node rank changes.

Figure 11 presents two subplots illustrating the impact of Sybil attacks on average energy consumption and average throughput for three routing algorithms: From the image below, we will identify the SFMRTS (dark blue bars), MRHOF-RPL (orange bars), and SecTrust (green bars). With the first use case, the Sybil attack, average energy consumption was found on SecTrust’s network, which turned out to be higher than those of SFMRTS and MRHOF-RPL. The note can be a conclusive indicator indicating that SecTrust is less energy-efficient than the other two during Sybil attack conditions In the second subplot, employed for receiving the average throughout Sybil attack, MRHOF-RPL shows higher throughput than SFMRTS and SecTrust. This result suggests that in either of all the conditions, the MRHOF-RPL mechanism might be able to provide a higher rate of data transmission as compared to any other network Through such evaluation, researchers can appreciate how these algorithms hold out against Sybil attack simulation, including the performance level their algorithms provide for maintaining network efficiency and security.

Figure 11 Sybil attack: average energy consumption and average throughput.

Conclusion

This study introduces the Metric-based RPL Trustworthiness Scheme (SF-MRTS) as an innovative routing system for RPL networks. It deeply emphasizes trust and cooperation; by simply deploying the multi-criteria-based trust metric ERNT, SF-MRTS helps optimize routing decisions at each hop along the path. Through simulations, we have demonstrated that SF-MRTS effectively reduces network security risks while maintaining high performance and stability. The results also indicate that the system’s low energy consumption and high packet delivery ratio resulted from the SF-MRTS’s capability to recognize and isolate attacks (black hole, Rank, and Sybil) and the energy-balanced topology mechanism. Energy and security are the two primary elements that this study is focusing on. In addition, we showed that ERNT meets the monotonic and isotonic characteristics criteria, allowing the SF-MRTS-based routing protocol to satisfy the consistency, optimality, and loop-freeness requirements. Additionally, we turned SF-MRTS into a tactic by using ideas from game theory. The SF-MRTS approach preserves the integrity of the network by punishing and isolating the uncooperative (i.e., untrusted) nodes, which forces nodes to cooperate rather than cheat to avoid being penalized. We found that the SF-MRTS approach is evolutionarily stable and that, given perfect monitoring, it is comparable to rivalry and spiteful approaches in terms of its capacity to encourage and enforce cooperation among nodes. This was demonstrated by our research into the collaborative creation of the SF-MRTS strategy.

Statement of Originality

I attest that the article I submitted to PeerJ Computer Science is my/our original work (with references to other literature included and cited in the correct format). In essence, the article addressed all the aspects of the declaration, including both text and figures, tables and data, and any extra content accompanying it.

Supplemental Information

Supplemental Information 1 Code

Supplemental Information 2 Detection prevention

Supplemental Information 3 A comprehensive resource for understanding the computational and analytical approaches undertaken in the trust task, providing insights into the methodologies and results that form the basis of the related contribution

Detailed analytical content related to the first contribution of a trust-related task. This includes code snippets, data analysis, and visualization segments that are integral to understanding the specifics of the trust task. Key components are: • Outlines the primary goals and objectives of the trust task being analyzed. • Steps taken to clean and preprocess the data used in the analysis. • Detailed visualizations and statistical summaries to explore the data’s characteristics. • Code and methodologies for implementing machine learning or statistical models related to trust metrics. • Analysis of the model’s performance, including accuracy, precision, recall, and other relevant metrics. • Summary of findings and potential implications of the analysis.

Additional Information and Declarations

Competing Interests

Author Contributions

Data Availability

The authors declare there are no competing interests.

Muhammad Zunnurain Hussain conceived and designed the experiments, performed the experiments, analyzed the data, performed the computation work, prepared figures and/or tables, and approved the final draft.

Zurina Mohd Hanapi conceived and designed the experiments, performed the experiments, analyzed the data, performed the computation work, authored or reviewed drafts of the article, and approved the final draft.

Azizol Abdullah conceived and designed the experiments, performed the experiments, analyzed the data, performed the computation work, authored or reviewed drafts of the article, and approved the final draft.

Masnida Hussin conceived and designed the experiments, performed the experiments, analyzed the data, performed the computation work, authored or reviewed drafts of the article, and approved the final draft.

Mohd Izuan Hafez Ninggal conceived and designed the experiments, performed the experiments, analyzed the data, performed the computation work, authored or reviewed drafts of the article, and approved the final draft.

The following information was supplied regarding data availability:

The code is available in the Supplemental Files.

The raw data is available at Zenodo: mzunnurainhussain. (2024). mzunnurainhussain/PhDContribution1: v1.1 (v1.1). Zenodo. https://doi.org/10.5281/zenodo.12748849.

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
