# Peer review of "An efficient secure and energy resilient trust-based system for detection and mitigation of sybil attack detection (SAN)"

_PeerJ Computer Science, doi:10.7717/peerj-cs.2231_

## Round 0.1 · original submission · Major Revisions

The paper introduces the Metric-based RPL Trustworthiness Scheme as a routing system for RPL networks, emphasizing trust and cooperation to optimize routing decisions. One positive aspect is its clear articulation of the proposed scheme's methodology and its relevance to the IoT network security domain. Additionally, the paper provides a detailed description of the experimental setup, which contributes to the replicability of the study. However, there are notable weaknesses in the paper's presentation. Several reviewers raised concerns about formatting issues, including blurry figures, lack of section numbering, and insufficient explanations of symbols and figures. These issues detract from the clarity and professionalism of the paper. Furthermore, the paper lacks thorough comparative analysis and interpretation of metrics, hindering the assessment of the proposed scheme's effectiveness against existing methods.

Reviewer 1 ·

Basic reporting

This paper has several formatting issues that need to be fixed

1. Please add hyper link of the references
2. all figures are unclear. please fix
3. proofreading is required for this manuscript, there are many grammatical issues.
For example, "We can see from the table that Sybil’s average values range from 140 to 180, with corresponding accuracy percentages varying between 0A scenario where a blockchain network is under siege." Not sure what is 0A scenario here.

Experimental design

The authors claim that they conduct experiments using simulation. However, the experimental design is very vague. It does not talk about how the data is simulated, which hinders my understanding of why the proposed method is correct.

Validity of the findings

As it's not clear how the authors simulate the data for experiments, it's hard to evaluate the findings

Additional comments

Thanks for submitting your work. I suggest that the authors fixed all the formatting issues with careful proofreading. The methodology of how the authors simulate data and conduct experiments is the major concern which hinders my assessment for this work. I suggest that the authors spend more efforts clearly explaining their experiments.

Reviewer 2 ·

Basic reporting

This study introduces the Metric-based RPL Trustworthiness Scheme (SF-MRTS) as an innovative routing system for RPL networks. SF-MRTS deeply emphasizes trust and cooperation; by simply deploying the multicriteria-based trust metric ERNT, SF-MRTS helps optimize routing decisions at each hop along the path. The authors demonstrates that SF-MRTS effectively reduces network security risks while maintaining high performance and stability.
The paper is bad at reporting. I list the problems as follows.
1)All the figures are blurred, and the readers can’t see any text in the figure.
2)The paper doesn’t have section number in the title of each section
3) There is no explanation of symbols in equations. For example, In Euqation1, what is Eeeg, Eam, f and r remains unknown.
4)Figure 2 crosses two pages and leaves large blank in page 8. Actually, the authors can resize it and put it in one page.
5) 2 bar charts take up 1 single page in page 14. The authors could merge all the bar chart into one chart which can be put into one page.
6) There are no explanations for Figure 7,8,9,10
7) The authors don’t put the essential information into the abstract and introductions but in the conclusion. In the abstract and introduction, the authors write too much about the background which should be few sentences but don’t clarify what the contribution of this paper is.

Experimental design

The results were averaged after conducting the simulations ten times with three different topologies which looks fair. But the authors don't clarify the research questions which is confusing to the readers.

Validity of the findings

The paper uses node rank changes, packet delivery ratio, energy consumption and throughput as metrics. The authors compare their method, SF-MRTS with MRHOF-RPL and SecTrust. However, the author never elaborates the meaning of all the metrics and it’s hard for the readers to figure out if their methods are better with confusion about the metrics. The authors only have 2-3 sentences about the compared methods in the literature review part and don’t demonstrate how their design can exceed the compared methods.

Reviewer 3 ·

Basic reporting

The paper is generally well-written, clear and unambiguous. The introduction and background provide sufficient context for the research. The structure mostly conforms to the standard. Some figures are not clear, such as, figure1, it is very hard to see the text within the figure clearly. The quality of figures should be improved.

Experimental design

The research focus on an efficient and secure trust-based system for detecting Sybil attacks in RPL-based IoT networks, which is well-definied, relevant and meaningful. The authors state how they address gaps in previous works. The method is described in detail.

Validity of the findings

The paper does not make strong claims about the impact or novelty of the work. The results demonstrate that the proposed SF-MRTS scheme is effective in detecting and mitigating Sybil attacks while maintaining network performance. However, some claims, such as the evolutionary stability of SF-MRTS, may require further justification and discussion of limitations.

---

## Round 0.2 · Minor Revisions

There are only a few minor issues with the figures, and some justifications are needed.

Reviewer 1 ·

Basic reporting

Authors fixed part formatting issues

Experimental design

Authors added detailed metrics used in the evaluation and how data is simulated

Validity of the findings

N/A

Additional comments

I want to thank the authors for their efforts for the revision. However, I still have several concerns regarding the manuscript.

[figure issues] while the authors claimed to have replaced the figures with high-resolution ones, it's still not clear enough. For example, words in figure 1 can barely be seen. Also, there are overlapping texts in figure 9 to be fixed. I wish the authors can spend more efforts for fixing those formatting issues.

[simulation scale] the authors mentioned that they used 20 nodes to simulate the behaviors. This number seems not scalable enough. And the authors did not make justification about why this scale is realistic in their scenario. Authors can consider conducting larger-scale experiments or justify for why their simulation setting is reasonable.

Reviewer 3 ·

Basic reporting

It answers my questions well in the response.

Experimental design

no comment

Validity of the findings

no comment

---

## Round 0.3 · accepted · Accept

As per the comment from Reviewer 1, please remove scalability consideration and Preliminary Results from Larger-Scale Simulations. Also, polish the figures.

Reviewer 1 ·

Basic reporting

N/A

Experimental design

N/A

Validity of the findings

N/A

Additional comments

thank the authors for fixing the issues. Just a few minor comments when you are preparing your camera-ready version.

1. In your submitted file, figure 1 is still blurry though the diff file did use a high-resolution figure. Please make sure to use the right figure for your final version.

2. I suggest the authors remove scalability consideration and Preliminary Results from Larger-Scale Simulations. I was initially expecting to see the authors' results on larger-scale data, but it seems they just stated that they did some experiments, but no results are included in this draft. If you don't have the plan to include those results, you can just move this part to future work.

Reviewer 3 ·

Basic reporting

no comment

Experimental design

no comment

Validity of the findings

no comment